# Assessment of the Antimalarial Treatment Failure in Ebonyi State, Southeast Nigeria

Chinedu Ogbonnia Egwu [1,*], Chinyere Aloke [1,2], Jennifer Chukwu [3], Joshua Chidiebere Nwankwo [4], Chinemerem Irem [4], Kingsley E. Nwagu [4], Felix Nwite [5], Anthony Ogbonnaya Agwu [5], Esther Alum [5], Christian E. Offor [5] and Nwogo Ajuka Obasi [1]

1  Department of Medical Biochemistry, College of Medicine, Alex Ekwueme Federal University Ndufu-Alike, Abakaliki PMB 1010, Nigeria
2  Protein Structure-Function and Research Unit, School of Molecular and Cell Biology, Faculty of Science, University of the Witwatersrand, Braamfontein, Johannesburg 2050, South Africa
3  World Health Organization, 4th Floor, United Nations House Plot 617/618 Central Area District, Abuja PMB 2861, Nigeria
4  Department of Biotechnology, Alex Ekwueme Federal University Ndufu-Alike, Abakaliki PMB 1010, Nigeria
5  Biochemistry Department, Faculty of Biological Sciences, Ebonyi State University, Abakaliki PMB 53, Nigeria
*  Correspondence: echojay2010@yahoo.com

**Abstract:** The fight against malaria is a continuum as the epidemic is not abating. For proper deployment of tools in the fight against malaria, an assessment of the situation is necessary. This work assessed the level of antimalarial drug treatment failure in Ebonyi State, Nigeria. Both survey and in vitro analyses were adopted. The survey was used to obtain qualitative information from both the malaria subjects and the pharmacies where antimalarial drugs are sourced. The results from the survey were complemented by an in vitro assay of the level of active pharmaceutical ingredients (APIs) in the commonly used artemisinin combination in Nigeria; artemether/lumefantrine. Results from the survey revealed that artemisinin combination therapies (ACTs) remain the mainstay in the treatment of malaria, even though other non-artemisinin drugs are still used. It also revealed that many patients still self-medicate, although, this may not be connected to the treatment failure seen among some malaria subjects. The in vitro assay showed that ACT contains the right quantity of APIs. Further surveillance is, therefore, necessary to understand the real cause of treatment failure among malaria subjects in Nigeria.

**Keywords:** malaria; survey; antimalarial; failure; ACTs; treatment

## 1. Introduction

Malaria remains a major public health concern in low- and middle-income countries (LMICs). Malaria accounts for at least 600,000 deaths annually [1]. Malaria is caused by *Plasmodium* parasites, which are spread to people through the bites of infected female *Anopheles* mosquitoes. There are several species of *Plasmodium*; however, *P. falciparum* is the most dominant in sub-Saharan Africa [2]. Access to antimalarial drug therapy and the growing resistance of malaria parasites to artemisinin and mosquitoes to insecticides are significant concerns in malaria control and elimination [3]. Early diagnosis and treatment with appropriate antimalarial drugs can prevent severe illness and lethal outcomes [4]. Therefore, it is crucial that the administered antimalarial drugs are of acceptable quality [5]. Assuring the quality of ACTs and other antimalarial drugs used to counter malaria is paramount in ensuring that the success of malaria prevention and control strategies is maintained. Worrisomely, reports show that fake drugs abound in malaria-endemic regions where 1 in every 10 drugs are said to be fake and one-third of antimalarial drugs from malaria-endemic countries failed chemical content analysis [6,7]. In addition to the use of antimalarial drugs, other approaches for malaria control are: the use of insecticide-treated

nets and outdoor and indoor insecticide spays [8]. Prevention remains the best control approach; however, for confirmed malaria cases, antimalarial drugs must be used [8].

At the patient level, poor-quality antimalarial drugs may result in treatment failure, leading to prolonged or severe illness and even death, as sub-therapeutic doses can increase the risk of recrudescence of malaria. At the provider level, this increases the burden on already limited resources and undermines confidence in health providers [9]. More so, the overuse, inadequate or incomplete treatment regimen, and counterfeit drugs, which lead to drug failures and the development of resistance by both the mosquitoes and malaria parasites to insecticides and antimalarial drugs, respectively, are also culpable [2,10,11]. From a public health perspective, drugs with low-stated active pharmaceutical ingredients (API) or low bioavailability may be selected for drug-resistant parasites [12].

A connection between the quality of artemisinin-based medicines and drug resistance has been postulated but not as yet established [13]. The increase in poor-quality (e.g., counterfeit or falsified) antimalarial drugs may cause an impediment to efficient malaria control [14]. Poor-quality antimalarial drugs have severe repercussions for public health [5]. Drugs with minute, or with no active pharmaceutical ingredients (APIs) may cause increased morbidity and death [15]. Moreover, little amounts of APIs in poor-quality drugs will result in sub-therapeutic concentrations of the drug in vivo, which may contribute to the selection of resistant parasites [16]. In addition, the use of poor-quality antimalarial drugs leads to economic loss for patients and their families, healthcare systems, and pharmaceutical companies producing the genuine product [17]. Information about the quality of antimalarial drugs is imperative for improving malaria treatment and effectively running malaria control programs [18]. The objective of this study is to study the failure rate of conventional antimalarial drugs, especially the artemisinin-based ones, among malaria subjects through survey and in vitro analysis. This study will provide information that is essential in policy making and implementation in the fight against malaria, especially in the best choice of antimalarial drug.

## 2. Materials and Methods

### 2.1. Study Area

A descriptive cross-sectional survey was carried out to access the failure rate of conventional antimalarial drugs, especially the Artemisinin-based ones, and evaluate the level of revisit to the pharmacies for the purchase of antimalarial drugs due to failure in selected major cities in Ebonyi state (Abakaliki, Afikpo, and Ikwo town).

### 2.2. Sampling

A sample size of 300 respondents was drawn from antimalarial drug users across major towns in Ebonyi state. Structured questionnaires were administered to the respondents to gather information on their drug use patterns. The respondents were given enough time to fill out the questionnaire and non-educated respondents were guided in filling out the questionnaires. The inclusion criteria were: respondents who have had malaria episodes in the last year; must live in urban areas of Ebonyi State and must be at least 15 years. The exclusion criteria were: respondents who took herbal medicine in the last six months and those less than 15 years.

Sixty-five (65) pharmacies across these major cities were also sampled. The antimalarial drug usage was ascertained from the pharmacies using structured questionnaires. The quality of drugs was assessed through the purchase of antimalarial drugs from a random sample of selected providers.

The responses from the respondents were analyzed using the sentiment levels and percentage levels of occurrence, respectively.

The sentiment levels of the respondents from the survey were taken and analyzed on a five-scale Likert Model as shown in Equation (1).

$$SL = \frac{1}{n}\sum(f \times SC) \tag{1}$$

SL = Average sentiment level; *n* = total number of respondents in the survey; f = frequency of respondents at each sentiment score; SC = sentiment score (1, 2, 3, 4, or 5).

The percentage level of occurrence was determined as shown in Equation (2).

$$level\ of\ occurrence = \frac{number\ of\ occurrence}{total\ number\ of\ occurrence} \times 100 \qquad (2)$$

### 2.3. Chemical Analysis of Sample (Antimalarial Drugs)

The gold standard antimalarial drugs (artemether/lumefantrine) in the formulation of 20/120 mg and 80/480 mg were purchased, respectively, from pharmacies sited in different cities of Ebonyi State, for the analysis. All the reagents and chemicals used for the laboratory analysis were of high analytical standard. The selected common antimalarial drugs were analyzed to ascertain the level of the active pharmaceutical ingredients (APIs) using thin-layer chromatography as described by Global Pharma Health Fund (GPHF) Minilab. Drug quality was assessed by comparing the amount of active ingredient in the eluents of each dissolution sample against a known concentration of the hypothetical standard for artemisinin combination therapy (ACT) after thin-layer chromatography. The hypothetical standard of 20/120 mg for single strength or 80/480 mg for double strength artemisinin/lumefantrine was used in order to rule out variations that may occur from any other standard drug from the field. The hypothetical standard is the recommended or ideal dose for artemether/lumefantrine. It was, therefore, used to establish the deviation of the analyzed samples from the recommended standard.

### 2.4. Ethical Consideration

Ethical approval was obtained from the Ethics Committee of Alex-Ekwueme Federal University Ndufu-Alike, Ebonyi State Nigeria with the reference FUNAI/R&D/2021/0036/Vol.2/00288.

### 2.5. Statistical Analysis

The data were analyzed using GraphPad Prism and the level of significance was tested at a 95% confidence interval.

## 3. Results and Discussion

### 3.1. Survey

The survey results gotten from the patients and pharmacy are presented.

#### 3.1.1. Patient Survey

- Periodicity of malaria and antimalarial drug use

Exactly 300 respondents (malaria subjects) were sampled in the study. All the respondents (*n* = 300) said that they have had malaria episodes before and where 46.33% of the respondents had malaria at least once in 6 months (Figure 1A). This finding corroborates the fact that Nigeria is a hyper-endemic area for malaria [19].

Our findings showed that about 35% of the malaria subjects still self-medicate when they have malaria whereas only about 20% of them visit a doctor for antimalarial drug recommendation (Figure 1B). About 80% of the malaria subjects prefer to source their antimalarial drugs from pharmacy stores while others get theirs from other sources such as caregivers, home nurses, etc. (Figure 1D). Self-medication is a common practice in developing countries. So many factors are responsible for the high level of self-medication in these regions. These factors may include but are not limited to the mildness of the disease; previous knowledge of the disease; unavailability of or poorly equipped health facilities, the unfriendly attitude of healthcare workers, and the high cost of hospitalization [20,21].

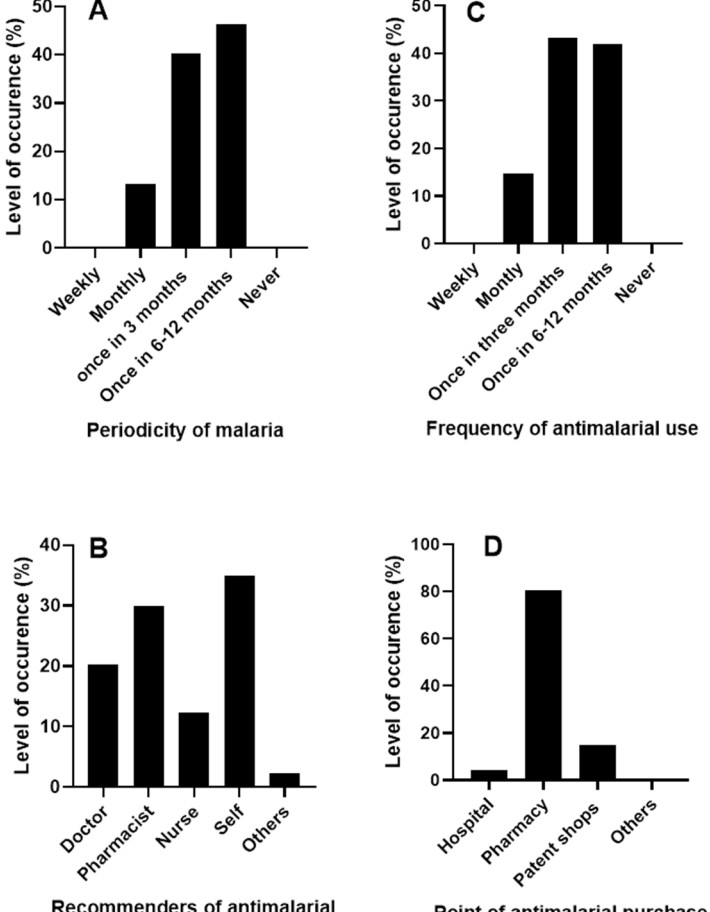

**Figure 1.** Periodicity of malaria. (**A**) Periodicity of malaria in Ebonyi State Nigeria (**B**) Recommenders of antimalarial drugs for malaria treatment (**C**) Frequency of antimalarial use in Ebonyi State Nigeria (**D**) Points of antimalarial purchase for treatment of malaria.

More so, the majority of the respondents (43.33%) (*n* = 130) use antimalarial drugs every three (3) months followed by those that use them once, in at least six months (Figure 1C). This finding shows that the relapse period is short. The majority of the antimalarials target the blood stages of the malaria parasites. However, some parasites have the ability to form dormant stages in the liver called hypnozoites which can trigger new episodes of malaria after a few months or years [22]. Even though this occurrence is common in *P. vivax* and not *P. falciparum* (the commonest species in Africa), epidemiological surveillance should look out for the presence of *P. vivax* in most African countries as malaria parasites are easily exported or imported [23].

- Choice of antimalarial in use and clearance of malaria signs and symptoms

Our finding shows that artemisinin-based combinations remain the most commonly used antimalarial in the treatment of malaria (90%) (Figure 2A). The choice of antimalarial is mostly predicated on the efficacy of the drug as indicated by the majority of the respondents (38.67% (*n* = 116) (Figure 2B). The efficacy of a drug is among the main factors to be considered in choosing a drug [24]. Artemisinin and its derivatives used in combination therapies as recommended by the WHO have remained the commonly used antimalarial due to its efficacy profile [1,25].

Generally, the symptoms of malaria are clear between 1–3 days (Figure 2C). The results also showed that 14.66% (*n* = 44) of the respondents feel signs and symptoms of malaria again after treatment in two weeks, 18.67% (*n* = 56) in 2–4 weeks, and 66.67% (*n* = 200) in >4 weeks (Figure 2D). The ACTs, which are the most commonly used antimalarials, drastically clear malaria parasites and resolve malaria symptoms such as fever fast [26].

However, a relapse of malaria in 4 weeks as suggested by most respondents in this study is a cause for worry. Quick relapse in malaria could be attributed to so many factors, which include but are not limited to resistance to antimalarial by the parasites, counterfeit drugs, and wrong diagnosis and treatment [2,10,11].

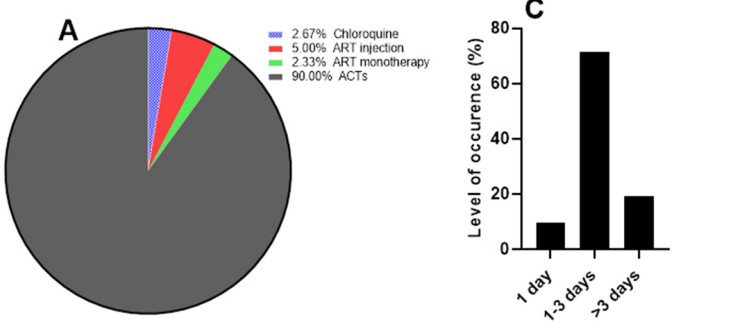

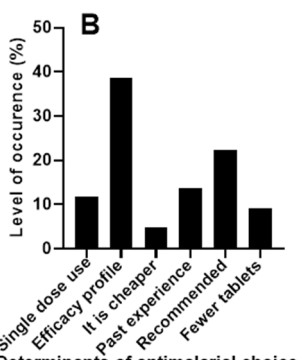

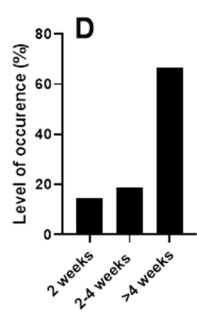

**Figure 2.** Choice of antimalarials. (**A**) Commonly used antimalarials by patients (**B**) Determinants of antimalarial choice (**C**) Time to clearance of malaria symptoms (**D**) Time to the reappearance of malaria symptoms after treatment.

● Patients' satisfaction with antimalarial drugs

The respondents seemed satisfied with the current antimalarials that they use as shown in Table 1. Analyzing the results in Table 1 using the Likert scale, the value falls at 4. This means that respondents generally feel satisfied with the antimalarial that they use. This satisfaction may be related to the comparative efficacy of ACTs over other antimalarials.

**Table 1.** Satisfaction level with antimalarials.

| Sentiment Level | Numerical Value | Response | Total |
|---|---|---|---|
| Highly Satisfactory | 5 | 92 | 460 |
| Satisfactory | 4 | 178 | 712 |
| Neutral | 3 | 22 | 66 |
| Unsatisfactory | 2 | 7 | 14 |
| Highly Unsatisfactory | 1 | 1 | 1 |
| Total | | 300 | 1253 |
| Average sentiment level | 4.18 | | |

3.1.2. Pharmacy Survey

- Antimalarials used for malaria treatment

From the pharmacy shops surveyed (*n* = 65) across the major cities of Ebonyi State, all the pharmacies sell ACTs (100%) (Figure 3A). The survey shows that ACTs account for the mostly purchased antimalarial drugs (98.46%) (Figure 3B). This corroborates the results from the patients' survey, which shows that ACTs are the most commonly used antimalarials (Figure 2B). This finding corroborates that of Mbah et al. who reported that ACTs are the most prescribed and sold antimalarial in Calabar, a city in South-southern part of Nigeria [27].

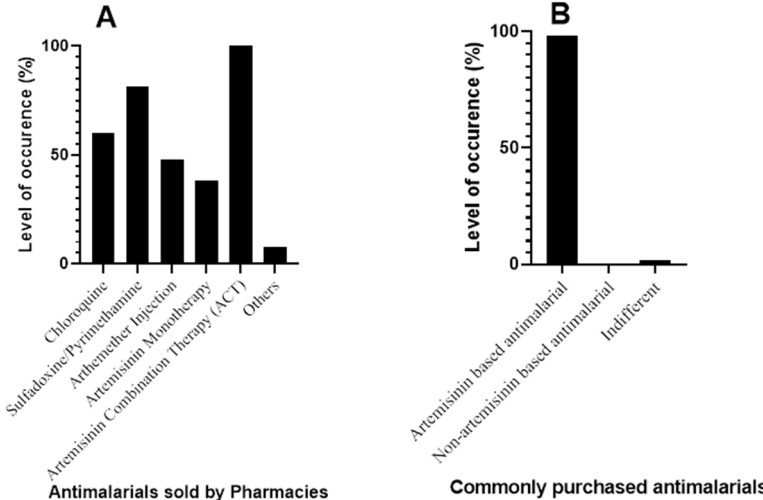

**Figure 3.** Antimalarials for malaria treatment—Pharmacy perspective. (**A**) Antimalarials sold by pharmacies (**B**) Commonly purchased antimalarials.

- Pattern of antimalarial drug prescription

Figure 4A shows that 52.31% (*n* = 34) pharmacies believed that the dry/rainy season affects the sale of antimalarial drugs while 47.69% (*n* = 31) believed it is not affected by the dry/rainy season. Furthermore, 70.77% (*n* = 46) pharmacies said that there was no remarkable change in the purchase pattern of antimalarial drugs while 29.33% (*n* = 19) pharmacies believed there was a change in the purchase pattern of antimalarial drugs across the year. Price change, season, epidemiology, non-adherence to preventive mechanisms, recrudescence, and reactivity on patients were among the causes of change in the pattern according to the respondent who believed that there was a change in pattern. Sub-Saharan Africa, where the burden is highest, has two major seasons annually—dry and rainy seasons; however, some areas may have a double-peaked rainy season. The transmission of the parasite, and by extension the disease outbreak, coincides with the spread of the rainy months in the different sub-Saharan African states [28–30]. During the season when there are increased pools of stagnant water, malaria episodes increase, consequently leading to increased sales of antimalarial during such seasons.

According to the survey of the pharmacies, about 43% of the pharmacies indicated that 25–50% of the patients present prescriptions before buying their antimalarials (Figure 4B). In developing countries, self-medication is still largely practiced for several reasons, which may include but are not limited to awareness of the disease, mildness of the disease, and lack of adequate health facilities [20]. This may have accounted for the low percentage of patients that buy antimalarials with prescriptions from physicians.

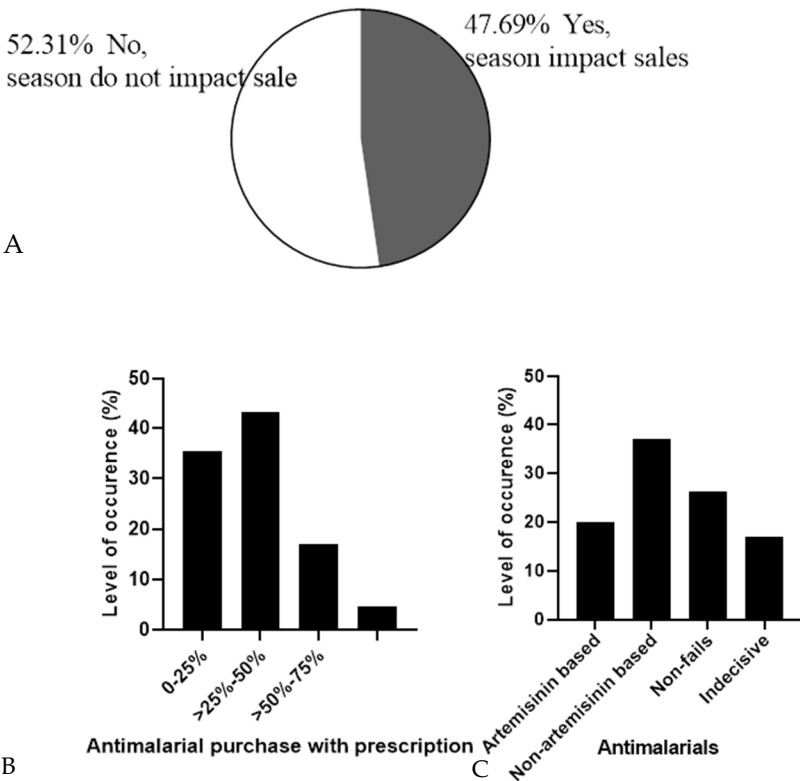

**Figure 4.** Pattern of antimalarial prescription. (**A**) Impact of rainy/dry season on antimalarial sale (**B**) Antimalarial purchase with prescription and (**C**) Failure pattern of antimalarials.

Considering the failure of antimalarials, 20% ($n$ = 13) pharmacies believed that ACTs mostly fail, and 36.92% ($n$ = 24) believed non-artemisinin-based antimalarials mostly fail. Exactly 26.15% ($n$ = 17) said that none of the antimalarials fail while 16.93% ($n$ = 11) were indifferent (Figure 4C). ACTs have been the recommended first-choice antimalarials by the WHO due to their efficacy standard [1]. The efficacy still recorded with other non-artemisinin-based antimalarials could be because of the regained efficacy of drugs like chloroquine against *Plasmodium* as seen in Malawi and other parts of Africa [31,32]. This, therefore, means that both artemisinin and non-artemisinin-based antimalarials are still essential in the treatment of malarial.

- Measurement of malaria relapse by sentiment level of pharmacies

The tendency of patients to come back for another treatment for malaria after the first treatment was measured using the sentiment level (SL) of the pharmacies from which the antimalarials are purchased and these were analyzed using the Likert scale as shown in Table 2. While some patients come back earlier than normal for another malaria treatment, others do not. The general feeling of the return for another antimalarial is undecided with an average SL of 2.86. As for whether relapse was higher in patients without a prescription, the pharmacies were undecided (SL = 2.83) and were also undecided about the contribution of purchase without prescription causing relapse (SL = 3.12). The cause of antimalarial treatment failure is multifactorial [33]. This may be why it was difficult for the pharmacies to emphatically link the relapse in malaria to treatment without a prescription.

**Table 2.** Measurement of malaria relapse by Sentiment level of pharmacies.

| Question | Strongly Agree | Agree | Undecided | Disagree | Strongly Disagree | Average Sentiment |
|---|---|---|---|---|---|---|
| Patients come back earlier than normal to buy antimalarial for treatment again due to failure of treatment? | 15 | 10 | 10 | 11 | 19 | 2.86 |
| Relapse is higher in patients without prescription | 10 | 8 | 22 | 11 | 14 | 2.83 |
| Antimalarial purchase without prescription contributes to treatment failure | 13 | 15 | 17 | 7 | 13 | 3.12 |
| Most patients buy antimalarial without diagnosis | 31 | 19 | 9 | 5 | 1 | 4.14 |
| Diagnosis before treatment can reduce treatment failure | 51 | 7 | 3 | 2 | 2 | 4.58 |
| Patients that have experienced treatment failure stick to the class of drug leading | 2 | 4 | 9 | 18 | 27 | 1.93 |

Note: Sentiment levels (SL)—Strongly agree = 5 (4.5–5), Agree = 4 (3.5–4.4), Undecided = 3 (2.5–3.4), Disagree = 2 (1.5–2.4) and Strongly disagree = 1 (0.5–1.4).

The pharmacies agree that patients buy antimalarials without a prescription (4.14). Even though most patients buy antimalarials without a prescription, the pharmacies believe strongly that diagnosis before treatment could reduce treatment failure (SL = 4.58). Prompt malaria diagnosis is essential for effective treatment especially in preventing the development of severe malaria and consequent death. In sub-Saharan Africa, rapid diagnostic tests (RDTs) are becoming increasingly the most used method to test for malaria [34]; however, the presumptive approach, which uses only signs and symptoms, is still largely practiced. Misdiagnosis can lead to the misapplication of antimalarials, which may pave way for the development of drug resistance and eventual treatment failure.

The pharmacies disagree that patients stick to their drugs after experiencing treatment failure (SL = 1.93). Every patient wants to get well as soon as possible while taking any drug. Failure in treatment after the use of an antimalarial, especially after rightful usage, may prompt a switch to other antimalarials. The WHO recommends a switch to another more effective first-line drug if the previous fails [35]. Several factors influence the choice of antimalarial use- duration of treatment, drug source, age, and income among other factors [36].

*3.2. Active Pharmaceutical Ingredient in Antimalarial Drugs*

To validate the results from both patients and pharmacy surveys, in vitro analysis was performed on some antimalarial drugs to ascertain their qualities. The samples were analyzed in comparison to a hypothetical standard based on the amount of active ingredients detected as shown in Figure 5. From the analysis, the 20/120 samples contained $22.15 \pm 3.1$ mg of artemether and $119.04 \pm 1.3$ mg of lumefantrine. On the other hand, the 80/480 samples contained $76.80 \pm 4.34$ mg of artemether and $453.20 \pm 27.68$ mg of lumefantrine. Analysis of the sample ($n = 10$) in comparison to a hypothetical standard, showed that there was no significant ($p > 0.05$) difference between the value of the sample

and the standard for artemisinin combination therapy (Hypothetical standard = 20/120 and 80/480, respectively). This implies that the antimalarial drugs met the European Pharmacopoeial requirements of 95–105% [37]. A drug contains excipients in addition to the APIs. These excipients can interact with APIs to enhance or compromise the activity of the APIs [38]. It is therefore a possibility that even when the APIs meet the recommended limit, the excipients may not. This can lead to compromised antimalarial activity. Surveillance of antimalarial APIs should also include the assay for types of excipients to ensure a holistic surveillance. This is especially important as the WHO says that 1 in every 10 medications in developing countries are counterfeits [6].

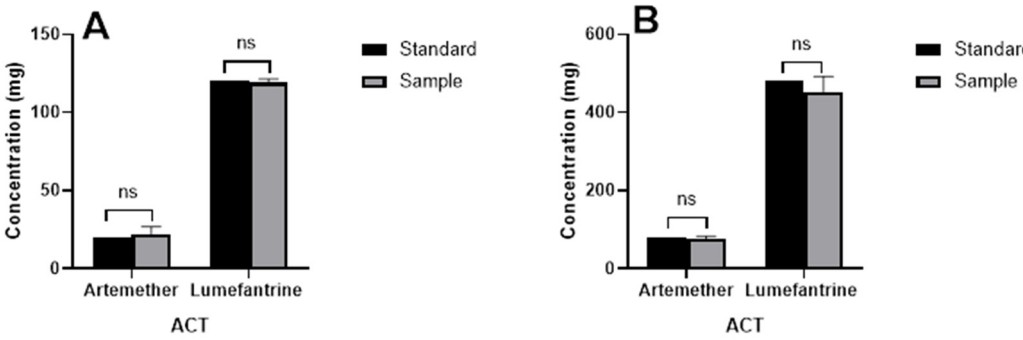

**Figure 5.** Active Pharmaceutical Ingredients in Artemether/Lumefantrine for (**A**) Single strength compared to the hypothetical standard 20/120 and (**B**) Double strength compared to the hypothetical standard 80/480. The values represent means from 10 samples each for 20/120 and 80/480 of artemether/lumefantrine respectively.

More so, failure from antimalarial treatment may result from resistance development from the malaria parasites which are evolutionarily dynamic. Drug resistance has almost become the fate of most antimalarial drugs [39]. Surveillance to understand the failure in malaria treatment should therefore include monitoring of the genotypic and phenotypic characteristics of the malaria parasites.

## 4. Conclusions

Our survey revealed that an artemisinin-based combination remains the gold standard in the treatment of malaria in Ebonyi state Nigeria. However, other classes of non-artemisinin-based therapy are still used to treat malaria. Our in vitro analysis showed that the commonest ACT in Nigeria, artemether/lumefantrine meets the right quality in terms of the level of active pharmaceutical ingredients. Failure to antimalarial treatment is multifactorial and may not be attributed to the level of active pharmaceutical ingredients in the drug. Regular surveillance of the APIs in addition to the level of excipients may be helpful in understanding the underlying causes of failure. Because of the limitations of the current study in terms of the number of respondents and the in vitro drug analysis, an expanded survey and drug analysis are recommended to fully understand the patterns in Nigeria.

**Author Contributions:** Conceptualization, C.O.E. and J.C.; methodology, J.C.N., A.O.A. and C.I.; software, C.O.E., J.C.N. and C.I.; validation, C.O.E., N.A.O. and C.E.O.; formal analysis, E.A.; investigation, J.C.N., C.I., F.N. and C.O.E.; resources, C.O.E., J.C.N., C.I. and A.O.A.; data curation, K.E.N. and C.O.E.; writing—original draft preparation, C.O.E., J.C.N., C.I., C.A., K.E.N. and N.A.O.; writing—review and editing, C.E.O., C.O.E., K.E.N., C.A. and N.A.O.; visualization, C.O.E.; supervision, C.O.E.; project administration, C.O.E. and J.C.N.; funding acquisition, C.O.E., J.C.N. and C.I. All authors have read and agreed to the published version of the manuscript.

**Funding:** This research received no external funding.

**Institutional Review Board Statement:** This work received ethical approval before execution from the ethical committee of Alex-Ekwueme Federal University Ndufu-Alike Ebonyi State Nigeria.

The study was conducted in accordance with the Declaration of Helsinki, and approved by the Institutional Review Board (or Ethics Committee) of Alex-Ekwueme Federal University Ndufu-Alike Ebonyi State Nigeria (FUNAI/R&D/2021/0036/Vol.2/00288 approved 26 July 2021).

**Informed Consent Statement:** Informed consent was obtained from all subjects involved in the study.

**Data Availability Statement:** Not applicable.

**Conflicts of Interest:** The authors declare no conflict of interest.

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
