# Peer review of "Assessment of the Antimalarial Treatment Failure in Ebonyi State, Southeast Nigeria"

_jox, doi:10.3390/jox13010003_

Round 1
Reviewer 1 Report
Well done study, merits publication.
Author Response
No comments from reviewer 1

Reviewer 2 Report
The article by Egwu et al. is a survey of malaria patients and pharmacies in Ebonyi State, Nigeria, with an aim to understand the probable reasons for the increasing failure of antimalarial treatments in the region. The survey provided insight into the infection rate and in the usage of antimalarial treatment methods prevailing amongst patients. The conclusions inferred from the survey are supported by the current data. However, with a sample size of 300 respondents, there should be caution in determining the general applicability of the conclusions. While the survey data presented in the manuscript is well-executed, the in vitro analyses of the antimalarial drugs for the presence of the active ingredients seem preliminary. The data is also not convincing enough to draw a conclusion. Therefore, the discussion of this data needs to be better written, including the need for additional validations. Given the public health burden of malaria, the article provides some important information that would be of interest to readers working towards implementation of effective malaria treatment methods. A few comments that would improve the manuscript are suggested below:
1. Abstract – The first mention of the acronym, ACT should be accompanied by the complete term.
2. Use of the term “antimalarial” should be specified as an antimalarial drug or measure throughout the manuscript as appropriate. E.g., Line 46.
3. Figure 1 title should be more descriptive about the data being presented in the panels.
4. Line 248 – How can a standard for quantification be “hypothetical”? It is unclear what the authors meant by this.
5. Line 254 - It is also unclear what the authors meant by hypothesized mean.
6. Figure 5 – The presentation of the data is confusing. What are the standards in this data? For the samples, it would be nice to show the individual sample measures alongside the means in the panels.
7. The authors should do a thorough check of the manuscript for grammatical errors and spellings.
Author Response
Response is attached

Reviewer 3 Report
The study of Chinedu O. Egwu aim the failure rate of conventional antimalarial especially the arteminin-based ones among malaria subjects through survey and in vitro analysis.
This is an important study in its context. But some remarks still remain:
Abstract
Abreviation of ACTs?
introduction
1. Reorganize this paragraph as:
Malaria remains a major public health 39 concern in low- and middle-income countries (LMICs). Malaria accounts for at least 600,000 deaths annually [1]. Malaria is caused by Plasmodium parasites which are spread to people through the bites of infected female Anopheles mosquitoes. There are several species of Plasmodium; however, P. falciparum is the most dominant in sub-Saharan Africa [2]. Access to antimalarial drug therapy and the growing resistance of malarial parasites to artemisinin and mosquitoes to insecticides, are significant concerns in malaria control and elimination [3]. Early diagnosis and treatment with appropriate antimalarial drugs can prevent severe illness and lethal outcome [4].
2. a brief discussion of the various malaria controls is needed to arrive at the drug strategy
Material and methods: you need to explain how you were able to measure certain parameters in this study
2.3. Chemical analysis of sample (antimalarial drugs): The gold standard antimalarials- artemether/lumefantrine in the formulation of 20/120 mg and 80/480 mg were purchased, respectively, from pharmacies sited in different cities of Ebonyi State, for the analysis.
I wish it had come from a firm and had had a quality control which really indicates the gold standard. If not, then what is the proof of their integrity?
results
figure 4 C : Failure pattern of antimalarials: How could you measure these parameters?
the sentiment level (SL): The general feeling of the return for another antimalarial is 218 undecided with an average SL of 2.86: How did you come up with this number?
3.2. Active pharmaceutical ingredient in antimalarial drugs: how many medicines were analysed in the end? Give the APIs per drug?
What about resistance genes in the parasite?
All in all, I'm not convinced of the in vitro analysis of ACT
